# Coalescence of multiple droplets induced by a constant DC electric field

Liwei Sun[1], Jian Wang[2], Yanhui Chen 🆔[1] *

1 School of Mechanical Engineering, Changchun Automobile Industry Institute, Changchun, China,
2 College of Computer Science and Technology, Jilin University, Changchun, China

* s7325894042022@163.com

Coalescence of multiple droplets induced by a
constant DC electric field. PLoS ONE 19(4):
e0300925. https://doi.org/10.1371/journal.
pone.0300925

Medicine, UNITED STATES

**Data Availability Statement:** All data is included in
the paper and/or Supporting Information.

**Funding:** The author(s) received no specific
funding for this work.

## Abstract

In this work, the electro-coalescence process of three nanodroplets under a constant DC electric field is investigated via molecular dynamics simulations (MD), aiming to explore the electric manipulation of multiple droplets coalescence on the molecular level. The symmetrical and asymmetrical dynamic evolutions of electrocoalescence process can be observed. Our MD simulations show that there are two types of critical electric fields to induce the special dynamics. The chain configuration can be formed, when one of the critical electric field is exceeded, referred to as $E_{cc}$. On the other hand, there is another critical electric field to change the coalescence pattern from complete coalescence to partial coalescence, the so-called $E_{cn}$. Finally, we find that the use of the pulsed DC electric field can overcome the drawbacks of the constant DC electric field in the crude oil industry, and the mechanisms behind the suppressed effect of the water chain or non-coalescence are further revealed.

## 1. Introduction

Recently, manipulation of dynamic characteristics of water droplets becomes to be an attractive topic due to its potentially promising in various practical applications, including fuel injection [1], ink-jet printing [2], aerosol delivery [3], lab-on-chip devices [4–6], electric dehydration [7–9], and so forth. A variety of external fields can manipulate the dynamic behavior of water droplets, for example, surface tension gradients [10, 11], thermal gradients [12], and approaches to an electric field [13–15]. Owing to the controllability and feasibility, electric manipulation of water droplets is widely adopted in recent years [16].

It is well known that, for the petroleum industry, coalescence of water droplets induced by an external electric field is one of the most effective ways to separate water from crude oil [17, 18]. When an electric field is applied, water droplets are polarized so that dipole-dipole force can be generated between polarized droplets. Consequently, adjacent droplets are forced to move towards each other by the resultant dipole-dipole force [19, 20]. During this period, the medium oil phase can be continuously drained by squeezing at the plateau border as droplets gradually approach. The continuous drainage of the medium oil phase makes the medium fluid thinner. There is a critical thickness of the film, below which the intermolecular force starts to play an important role. The attractive van der Waals force promotes droplet

**Competing interests:** The authors declare that they have no known competing financial interests or personal relationships that could have appeared to influence the work reported in this paper.

coalescence while the double-layer repulsion prevents this procedure. The film subsequently develops to be a metastable state when the mentioned two molecular forces do balance each other. Eventually, the film can be ruptured by the dipole-dipole force, and subsequent a liquid bridge is formed to link two adjacent droplets. The unbalanced pressure difference drives water to flow into the liquid bridge, and the coalescing droplet can be formed after process of liquid bridge expansion is achieved [16].

The direct constant (DC) electric field is found to be very effectively to promote process of the electro-coalescence, owing to its continuously high electric field strength. Additionally, the relationship between the strength, $E$, of the DC electric field and the electrocoalescence rate had been discussed in Ref. 21. The authors found that the coalescence rate is proportional to $E$ over a relatively low electric field range, whereas the rate of electrocoalescence is proportional to $E^2$ when the electric field strength is relatively high [21]. Although, the increasing field strength can accelerate the electrocoalescence process, whereas the liquid bridge linking water droplets could be destroyed to form non-coalescence pattern once the applied electric field exceeds a critical value, $E_{cn}$. Using molecular dynamics simulation (MD), Wang et al. [22] observed the dynamic process of complete coalescence, partial coalescence, and non-coalescence on the molecular level. By tracking movement of dissolved ions, they confirmed that partial coalescence and non-coalescence can be ascribed to the ions exchange through the liquid bridge. The radius of droplets is another important parameter affecting the electro-coalescence process, and the increasing radius can result in a faster coalescence process and a lower value of the critical electric field. It is due to the fact that droplets with large radii may dissolve a much larger number of salt ions that can increase electrostatic force and promote the ion exchange. The partial coalescence and non-coalescence can procedure some tiny droplets (the so-called secondary droplets), whose volume is very small compared to the primary droplets. These tiny droplets are difficult to be removed from the crude oil, which is undesirable in demulsification. Additionally, electric field type and electric field waveform are two important characteristic parameters in the usual electro-coalescence process. The strength of critical field can be greatly increased when the constant DC electric field is replaced by a pulsed DC one. Therefore, the pulsed DC electric field has a better performance on suppressing the non-coalescence behavior [23–25]. The influence of electric field waveform on the dynamic electrocoalescence had been investigated in previous investigation [26]. The authors noted that triangular waveform is the most effective to suppress the non-coalescence, and sinusoidal waveform takes the second place, whereas applied square wave is easily to trigger the non-coalescence. When a DC electric field is applied to the water in oil system, water droplets would easily link up into a stable chain, which not only reduces coalescence rate but also leads to short-circuit. The water chain is considered to be one of the major retarding factors in the crude oil industry [19]. Scientists found that the pulsed DC electric field can effectively avoid formation of water chains, whereas, the mechanisms behind the water chain are still very scant [23].

Previous studies mainly focused on electrocoalescence of the binary droplets system, and the dynamic behavior had been well investigated through experimental and theoretical analysis [17, 20, 27]. Although some important conclusions had been drawn to provide guidance for the practical application, whereas for crude oil industry, the system generally contains multiple water droplets. Up till now, natural insights into the dynamic coalescence of multiple droplets are still very poor. In this work, the electro-coalescence of multiple nanodroplets with dissolved salt ions is investigated via MD simulations. Different behavior can be observed during coalescence process, including complete coalescence, forming water chain configuration, and bouncing off droplets. The precondition for chain configuration and behavior of non-coalescence are investigated and compared. Finally, we find that the pulsed DC electric field not only can suppress water chain but also prevent non-coalescence behavior from occurring.

## 2. Simulation model and method

For the present work, we perform MD simulations to investigate dynamic evolution of coalescence process of three nanodroplets induced by external electric fields. The initial system contains three charged nanodroplets, located at the left, central, and right sides, as shown schematically in **Fig 1**. It is worth noting that the number of water molecules is directly related to the computational efficiency. Previous studies frequently simulated electro-coalescence process of nanodroplets with 3360 water molecules, which has a high computational efficiency [23, 27]. Therefore, we select the same number of water molecules for investigating the electro-coalescence process of multiple droplets system, and the corresponding radius is 3 nm. These three droplets are separated by the same gap thickness of $l_1 = l_2 = 3$ nm, and the corresponding centroid distance between two adjacent droplets is $l_c = 9$ nm. The simulated domain is a vacuum box with dimensions of 24 ×27× 24 nm$^3$, and we select periodic boundary conditions in $x$-, $y$-, and $z$-direction. The ion content has an important effect on the electrocoalescence behavior of nanodroplets. For convenience, each water droplet dissolves 120 KCl ions and the corresponding concentration is 1.32 M KCl, as that for Ref. 9, which is an appropriate concentration for investigating the electrocolaescence process. Here, the charge of K atom is +1, while it of Cl atom is -1.

The simple point charge/extension (SPC/E) model [28–30] is employed to investigate the dynamic coalescence process induced by an electric field, owing to its suitable capture of various liquid water properties in MD simulations. The water-water and ion-ion interactions are computed through pairwise potentials as a sum of Lennard-Jones (LJ) and electrostatic

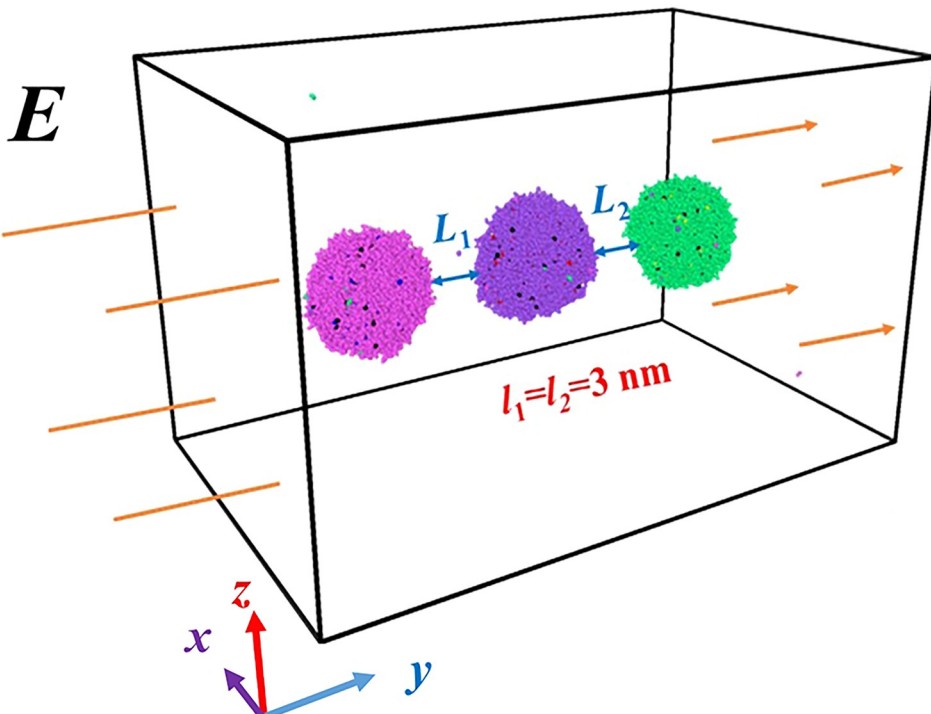

**Fig 1. The schematic of the simulated system, containing three charged water nanodroplets with dissolved ions (KCl).** The constant DC electric field along the negative y direction is applied to induce the droplet coalescence. Here, the initial gap thickness between separating droplets has the same value of $l_1 = l_2 = 3$ nm.

contributions, which is expressed as

$$U_{ij} = \frac{q_i q_j}{r_{ij}} + 4\varepsilon_{ij}\left[\left(\frac{\sigma_{ij}}{r_{ij}}\right)^{12} - \left(\frac{\sigma_{ij}}{r_{ij}}\right)^{6}\right]\left(r_{ij} < 10 \text{ Å}\right) \tag{1}$$

where $q_i$ and $q_j$ represent the partial charges of atoms $i$ and $j$, $r$ is the distance between these two atoms, $\sigma_{ij}$ is the zero-crossing distance, and $\varepsilon$ is the minimum energy parameter. The parameters of L-J potentials were validated in Refs. [31, 32]. The L-J and electrostatic interactions are spherically truncated with a cutoff of 10 Å. The long-range contribution of electrostatic interactions is solved using the particle-particle-particle-mesh (PPPM) method with a precision tolerance of $10^{-4}$ for the force calculation in the reciprocal space [33]. It should be noted that the SPC/E water model is non-polarisable; therefore, the change of orientation of water molecules is the main effect of electric field. In the SPC/E model, the charge of H atom is +0.4238, while it of O atom is -0.8476.

After building the initial system, we perform the MD simulations through the LAMMPS software package in the NVT ensembles (canonical ensemble) with a time step of 1 fs to reach an equilibrium state. During this period, the system is coupled to a Nose-Hoover thermostat to maintain the temperature at 298 K [34, 35]. The temporal evolution of the simulated system is conducted by solving Newton's equations of motion using the velocity-Verlet integrator with a simulation time step of 1 fs. The mass center of each droplet is fixed at its initial position by removing the mass center velocity. For the electrocoalescence step, the constant DC electric field, pointing from left to right, is applied to the system, and an additional force $F_i = q_i E$ is imposed on each particle in the presence of the electric field.

For the real electrocoalescence system, previous experiment had demonstrated that complete coalescence occurs under the action of a low electric field strength, whereas droplets may bounce off soon after they contact with each other by the presence of a high strength of electric field [21]. For MD simulations, it is very time-consuming if we simulate a real water-in-oil system. Therefore, we simulate the electro-coalescence process in the vacuum to save the computational cost, which is adopted in previous work [16]. The simulated results for complete coalescence and non-coalescence are compared to experimental observation from a previous study [36]. As shown in **Fig 2**, simulated results show a good agreement with the experimental results.

## 3. Results and discussion

### 3.1 Dynamic electro-coalescence process under DC electric field

When a constant DC electric field with $E = 0.3$ V nm$^{-1}$ is applied, the dynamic electrocoalescence process of three droplets is shown in **Fig 3**. At the moment of $t = 0$ ps, without the applying electric field, these nanodroplets are spherical owing to the surface energy minimization. Water molecules are subsequently polarized in the electric field circumstance, leading to the realignment of polarized molecules. Almost all atoms within water droplets yield in such a way, that electropositive hydrogen atoms are oriented toward the electric field direction whereas electronegative oxygen atoms are against the electric field direction. As a result, water droplets, acting as the induced dipole, could generate an electrostatic force to make separating droplets approach each other in opposite directions, and be elongated to the ellipsoidal shape by $F_i$, as shown at $t = 45$ ps. The snapshots also show that driven by the electrostatic force, droplets located on two sides move towards the central droplets. The left droplet first contacts with the central droplet at $t = 146$ ps, and the time requirement from the initial position to droplet contact here is defined as the contact time $t_c$, so $t_c = 46$ ps. Soon after that, the contact

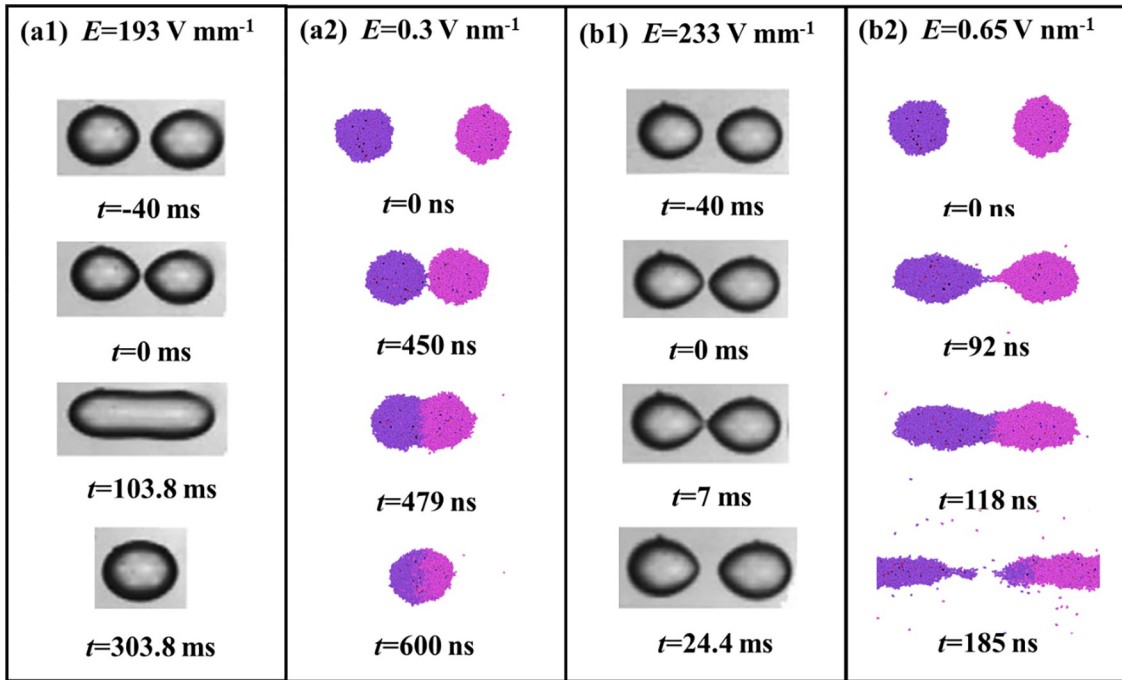

**Fig 2.** Experimental results of complete coalescence at (a1) $E$ = 193 V mm$^{-1}$ and non-coalescence at (b1) $E$ = 233 V mm$^{-1}$; MD results of complete coalescence at (a2) $E$ = 0.3 V nm$^{-1}$ and non-coalescence at (b2) $E$ = 0.65 V nm$^{-1}$.

between the right and central droplets occurs at $t_c$ = 150 ps, very similar to the mentioned contact process. Liquid bridges are then formed to link these droplets, and water molecules would flow into the liquid bridge resulting in the growth of such liquid bridges. The separating droplets then evolve into a big coalescing one at $t$ = 190 ps when the width of the liquid bridge equals that of the coalescing droplet. To form the energy minimization configuration, the retraction of the coalescing droplet takes place, and its shape changes to a stable ellipsoid when the additional force ($F_i$) and the surface force ($F_s$) do balance each other, as shown at $t$ = 1000 ps.

A dimensionless length of $L = l/r_0$ is here employed to obtain the variation of the gap thickness between adjacent droplets before droplets contact, as shown in **Fig 4A**. The decline of the dimensionless length in **Fig 4A** indicates the separating distance is decreased as droplets approach. Two curves are observed to be overlapped indicative of the approach degree of the left and right droplets to the central one being the nearly during droplets approach. Additionally, the dimensionless length displays two different decline rates. The dimensionless length reduces linearly when the gap thickness is large, while the rapid decrease of $L$ is observed with a parabolic morphology after $t>110$ ps. When the gap thickness is larger, the droplets' approach only depends on the electrostatic force. While the intermolecular force would be introduced to accelerate the droplet approach when the gap thickness reduces to less than the cutoff distance, so as to a rapid decrease of $L$. The variation of the centroid coordinates along the $y$-direction during droplets coalescence is also obtained to further understand the coalescence dynamics, as shown in **Fig 4B**. It is clearly demonstrated that the variation of droplets mass center located at two sides is completely symmetric about the central droplet (which is immobile with the invariable value of the centroid coordinate), refers to as the symmetric coalescence dynamics. The centroid coordinate of the left/right droplet initially changes slowly ($<110$ ps), whose value is increased for the left droplet and decreased for the right droplet.

## Symmetrical coalescence process

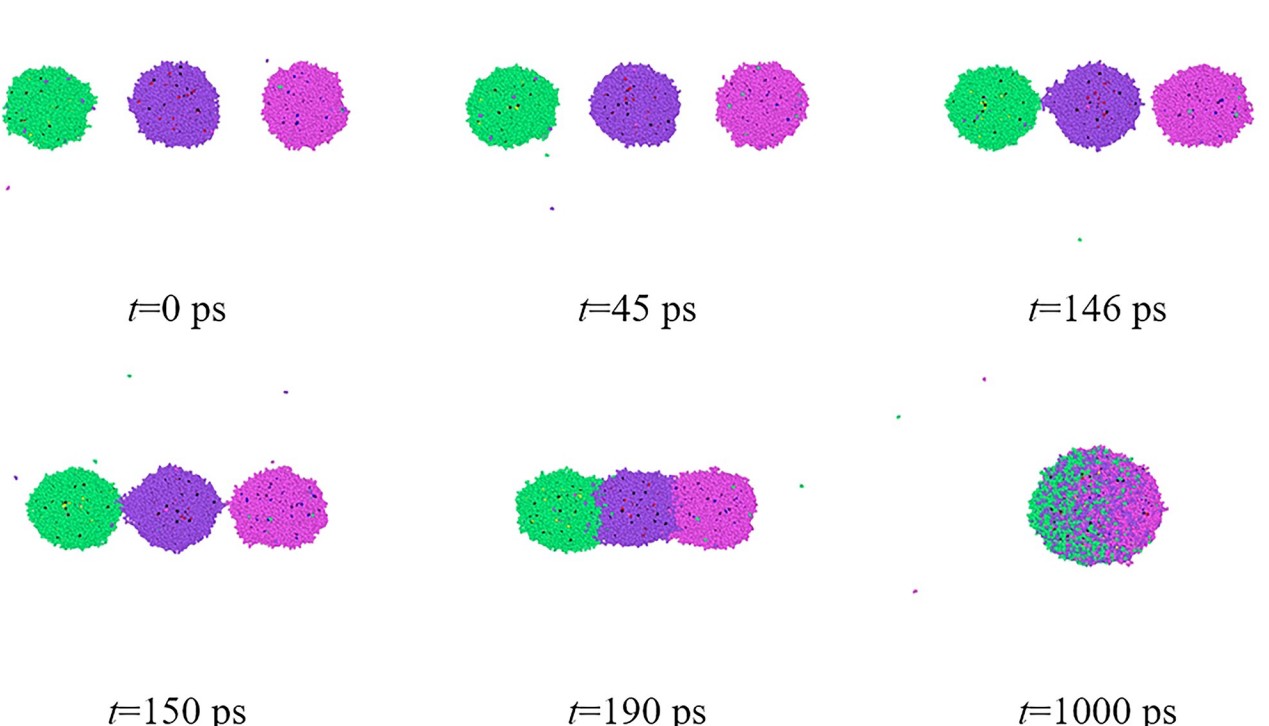

**Fig 3. Snapshots of dynamic coalescence process of three charged nanodroplets under a constant DC electric field of $E$ = 0.3 V nm$^{-1}$.**

After 110 ps, the intermolecular force accelerates the droplet approach so as to a fast change of the centroid coordinate. The variation rate of the centroid coordinate goes down again when complete coalescence occurs and the centroid coordinate of each droplet finally stabilizes at a stable value.

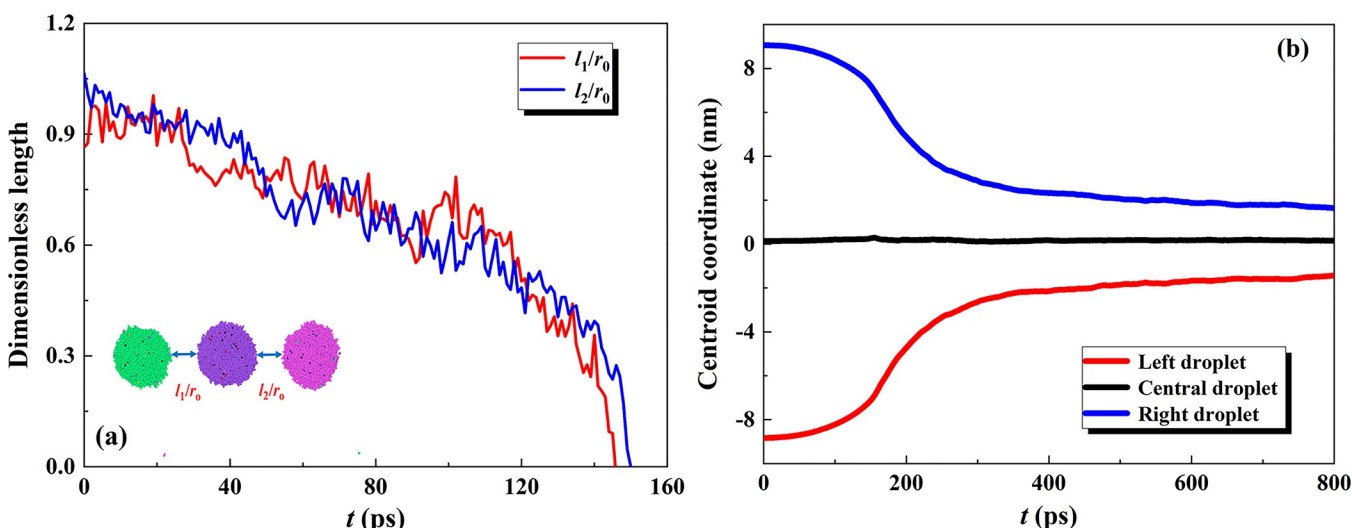

**Fig 4.** Variation of (a) dimensionless length before droplets contact and (b) centroid coordinate during coalescence process as a function of time under $E$ = 0.3 V nm$^{-1}$.

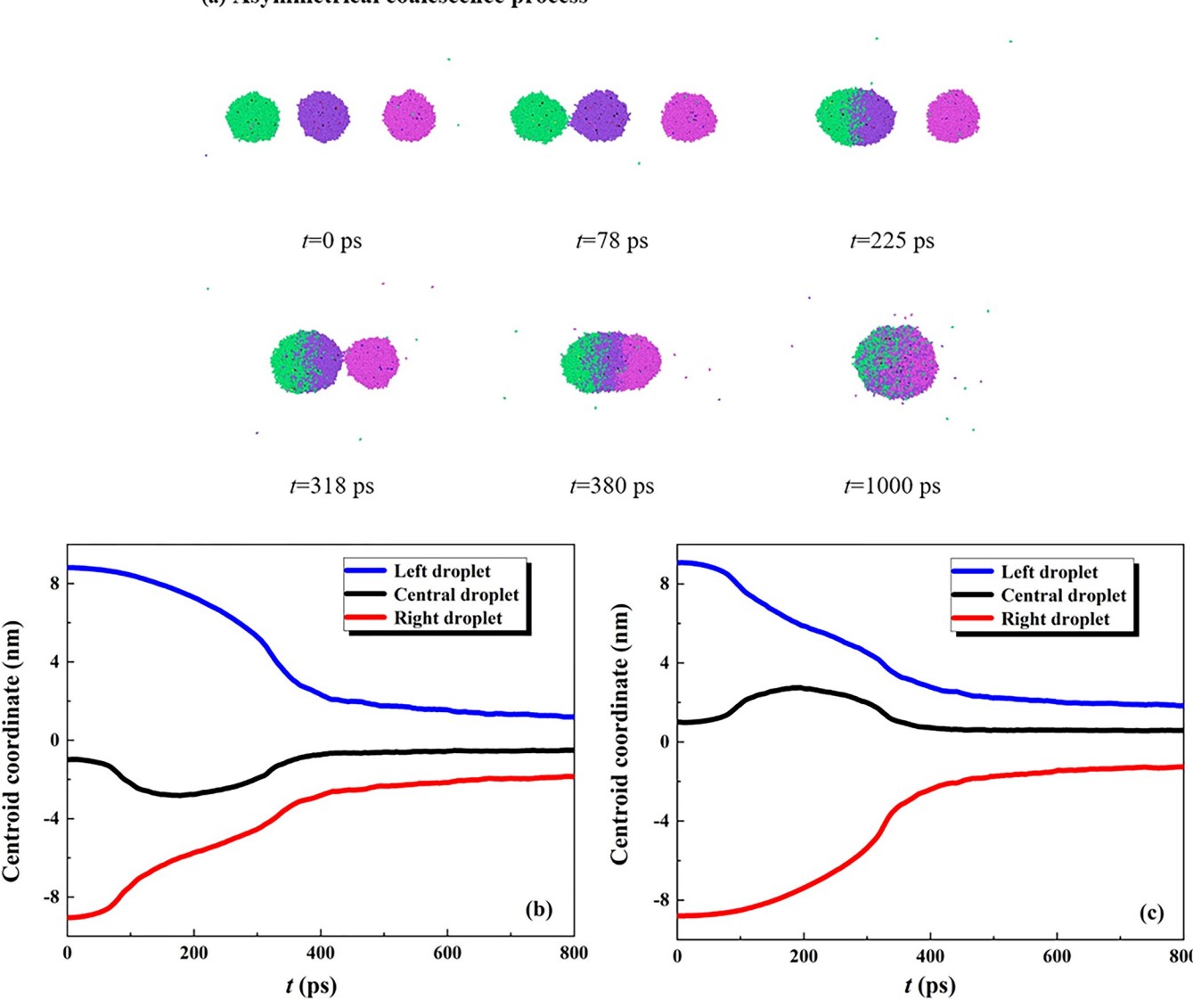

**Fig 5.** (a) The asymmetrical coalescence dynamics of three charged droplets under $E = 0.3$ V nm$^{-1}$. Here, the initial gap thicknesses possess different values of $l_1 = 2$ nm$< l_2 = 4$ nm. Variation of centroid coordinate during coalescence for (b) $l_1 = 2$ nm$< l_2 = 4$ nm, and (c) $l_1 = 4$ nm$> l_2 = 2$ nm.

We then establish the initial system with different gap thicknesses between separating droplets to investigate the dynamic coalescence procedure. The gap thickness $l_1$ is reduced to a value of 2 nm and the corresponding $l_2$ is increased to 4 nm ($l_1 < l_2$), and the dynamic evolution of the electro-coalescence process under $E = 0.3$ V nm$^{-1}$ is shown in **Fig 5A**. Previous studies frequently demonstrated that the electrostatic interaction between adjacent droplets ($F_e$) is a function of the electric field strength and gap thicknesses, which should be increased with the increase in the strength or decrease in the thicknesses. For the coalescence case in **Fig 5A**, the electrostatic interaction between the left and central droplets ($F_{e-l-c}$) is of course larger than that for the $F_{e-r-c}$. Hence, the symmetrical dynamics are thus broken which is accompanied by coalescence of the left and the central droplets should occur in the first step. The variation of the centroid coordinates in **Fig 5B** shows that the central droplet, not being immobile, starts to move towards the left droplet due to the existing unbalance electrostatic

interaction. The left droplet contacts with the central one at $t$ = 78 ps, and then coalesce into the first coalescing droplet with a quick change in their center-of-mass coordinate (78 ps$<t<$180 ps in **Fig 5B**). The droplet located at the right side persistently move to the central droplet from being to the end and contacts with the first coalescing droplet at $t$ = 318 ps to form the second coalescing droplet after 380 ps. The asymmetrical dynamics can also be observed in the other scenery ($l_1$ = 4 nm$>l_2$ = 2 nm), and the corresponding variation of the mass center is shown in **Fig 5C**.

## 3.2 Etrocoalescence procedure under various electric fields

The dynamic evolution of the multiple droplet system shows that, after three droplets contact, they would involved in an elongated droplet with some certain deformation degree under the action of the electric field force. Eow et al [37] experimentally investigated the topological evolution of a single aqueous drop under a constant DC electric field. The dimensionless deformation ratio, $D = d_1-d_2/d_1+d_2$, was proposed in their work to evaluate the deformation degree, where $d_1$ and $d_2$ are the long axes and the short axis of the droplet, respectively. They found that the deformation degree of the droplet and the associated $D$ would be significantly increased with the increase in electric field strength with its shape evolving into the spindle shape from the sphere. Such a droplet would be split to form some secondary droplets after the electric field exceeds its critical value. Therefore, the deformation degree of the droplet is an important index factor in electric-field-control droplet dynamics, which is also investigated in this work. We then select the dimensionless deformation ratio ($D$) to quantitatively analyze the deformation degree, and the dimensionless deformation ratio versus the increasing electric field strength, ranging from a low electric field of $E$ = 0.1 V nm$^{-1}$ to a relatively high value of 0.38 V nm$^{-1}$, is shown in **Fig 6A**, and some representative configuration of the deforming droplet can be observed in **Fig 6B**. According to the definition of the deformation ratio, there are two limits occurring under two extreme conditions. One is $D$ = 0, indicating the droplet can maintain a spherical shape without any deformation. The other extreme value is $D$ = 1, representing that the droplet is elongated to a line-like droplet with infinitely thin. When a low electric field ranging $E<$0.15 V nm$^{-1}$ is applied, the droplet shape only depends on the surface force ($D$ has a very low value $<$0.013) so that the water droplet could maintain a spherical morphology with negligible deformation. When $E>$0.15 V nm$^{-1}$, electric field force is indispensable status in determining the morphology of the coalescing droplet, and the stable elongated droplet is indicative of the balance of the surface force and the electric field force. For applying an intermediate electric field strength, 0.175 V nm$^{-1}<E<$0.325 V nm$^{-1}$, a more pronounced deformation of the coalescing droplet in **Fig 6B** can be observed, and $D$ in such situations increases linearly with the increase in the electric field strength. At high electric fields $E>$0.325 V nm$^{-1}$, a fast increase of $D$ is exhibited in **Fig 6A**, and maintaining a prolate spheroid gradually being arduous work. After that, we continuously increase the electric field strength to 0.4 V nm$^{-1}$ to observe the electro-coalescence process at such a high electric field and the associated deformation of the coalescing droplet, and the evolution of the electro-coalescence case is shown in **Fig 7**. **Fig 7** shows that droplets located on two sides can be observed to quickly move toward the central droplet and forms a coalescing droplet (150 ps). We find that the surface force cannot be balanced with the electric field force so the coalescing droplet is suffered from being elongated by the additional force ($F_i$) together with the electrostatic force ($F_e$) stemming from the adjacent simulated domain under the periodic boundary conditions. Eventually, the water droplet can be elongated to a water line throughout the whole domain and connects with liquid water from the adjacent simulated domain, the so-called stable water

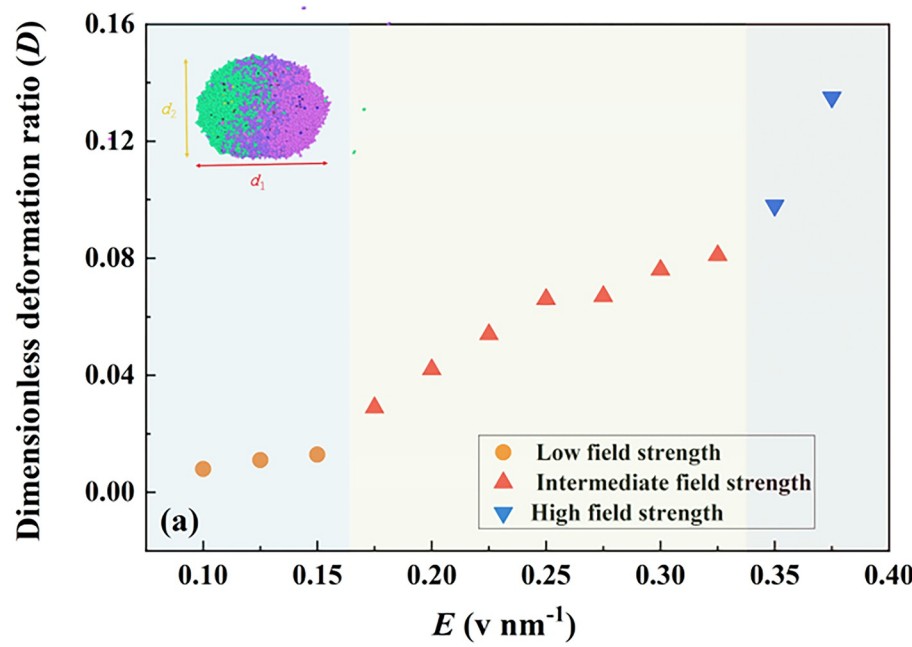

**(b)**

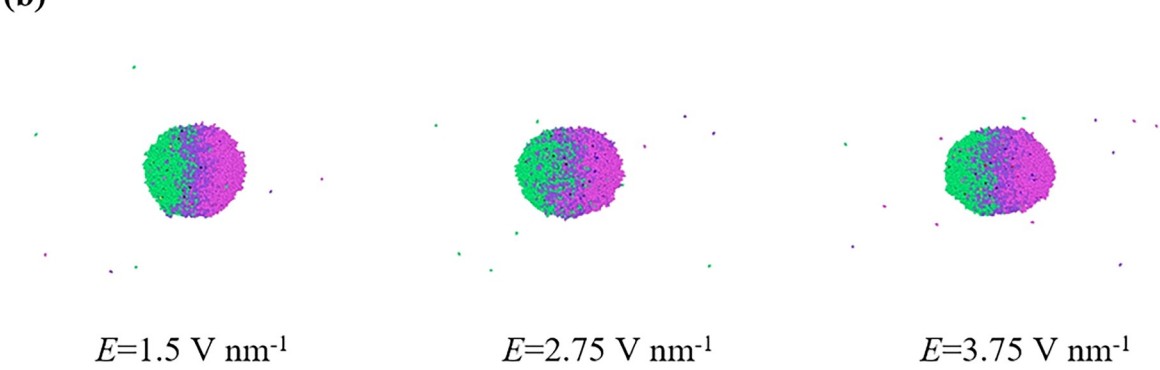

$E$=1.5 V nm$^{-1}$ $E$=2.75 V nm$^{-1}$ $E$=3.75 V nm$^{-1}$

**(c)**

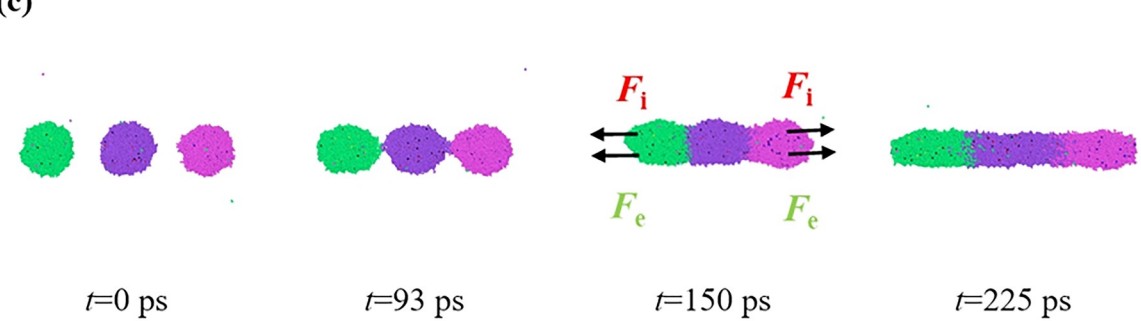

$t$=0 ps $t$=93 ps $t$=150 ps $t$=225 ps

**Fig 6.** (a) Variation of the dimensionless deformation ratio versus the increasing electric field strength ranging from 1.0 to 3.75 V nm$^{-1}$; (b) the deformation of the coalescing droplet under $E$ = 1.5, 2.75, and 3.75 V nm$^{-1}$; (c) forming the chain configuration under $E$ = 4.0 V nm$^{-1}$.

**(a)**

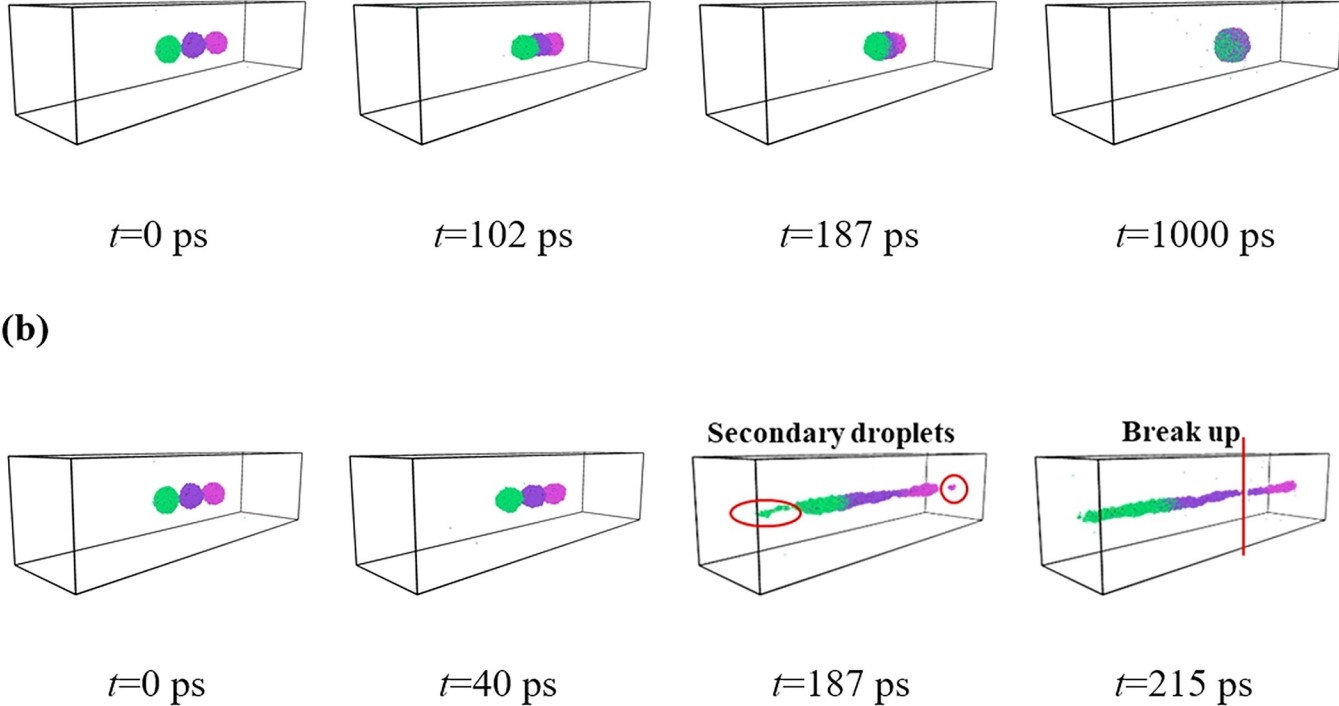

**(b)**

**Fig 7.** (a) Dynamic coalescence process under $E = 0.4$ V nm$^{-1}$ within an enlarging simulation domain, and (b) breakup of the coalescing droplet under $E = 0.51$ V nm$^{-1}$, leading to the non-coalescence dynamics.

chain. Therefore, the critical electric field strength for inducing the chain configuration is thus $E_{cc} = 0.40$ V nm$^{-1}$, and the corresponding critical deformation ratio is $D_c = 0.135$.

We then enlarge the space of the simulated domain along the $y$ direction to a value of 77 nm (dimensions of the other two directions are fixed at their initial values) to investigate whether the enlarging simulated domain would affect the electro-coalescence dynamics. The dynamic evolution of the electric coalescence process under $E = 0.4$ V nm$^{-1}$ is shown in **Fig 7A**. As shown in **Fig 7A**, owing to the separating distance of $l_1 = l_2 = 3$ nm, symmetrical coalescence dynamics is thus exhibited regardless of the enlarging simulated domain. We observe that the final droplet is in a spheroidal shape rather than the chain configuration, as shown at $t = 1000$ ps. The different dynamic behavior is a result of the electrohydrodynamic. In our MD simulations, periodic boundary conditions are applied in $x$-, $y$-, and $z$- directions, indicating that the left nanodroplet can interact with the right one induced by van der Waals force together with electrostatic force if two droplets are separated by a relatively small distance. Therefore, the left droplet can be elongated to contact with the right one to form a long water chain when the simulated domain is enough small. However, the intermolecular and electrostatic forces are significantly reduced with increase in the simulated domain, and hence the surface force plays an important role in controlling the shape of the coalescing droplet with involving into a spherical shape. With the progressive increase in the field strengths, we expect to explore the critical electric field for triggering the chain configuration. The electric coalescence procedure of three nanodroplets with an enlarging domain under $E = 0.51$ V nm$^{-1}$ is

exhibited in **Fig 7B**. A rapid symmetrical coalescence is shown with the contact time being approximately $t_c = 40$ ps. The electric field force imposed on two sides of the coalescing droplet is sufficiently too high to stretch the droplet into the spindle shape. The coalescing droplet cannot maintain an entire one with the secondary droplet first ejection from the primary droplet, as shown at $t = 187$ ps. This is explained in the previous study, secondary droplets with dissolved ions can be accelerated with more kinetic energy to overcome the constraint of the coalescing droplet [24]. Interestingly, we find that the secondary droplet at the left side contains more number of water molecules compared with that for the right side. To explain this, we calculate hydration number at the time point when the secondary droplets generate to estimate stability of hydration effect, which is defined as the average number of water molecules in the first solvation shell, expressed as follow:

$$N_{\text{ion}-O}^{\text{sol}} = \rho_{\text{ion}} 4\pi \int_0^{r_{\text{sol}}} g_{\text{ion}-O}(r) r^2 \mathrm{d}r \qquad (2)$$

where $\rho_{\text{ion}}$ is the number density of ions, $g_{\text{ion-O}}(r)$ is the ion radial distribution function, $r_{\text{sol}}$ is the radius of the first solvation shell corresponding to the first valley in curve of $g_{\text{ion-O}}(r)$. As shown in **Fig 8**, the first valley of $g(r)$ appears at 3.45 Å for K$^+$–O and at 3.95 Å for Cl$^-$O. It

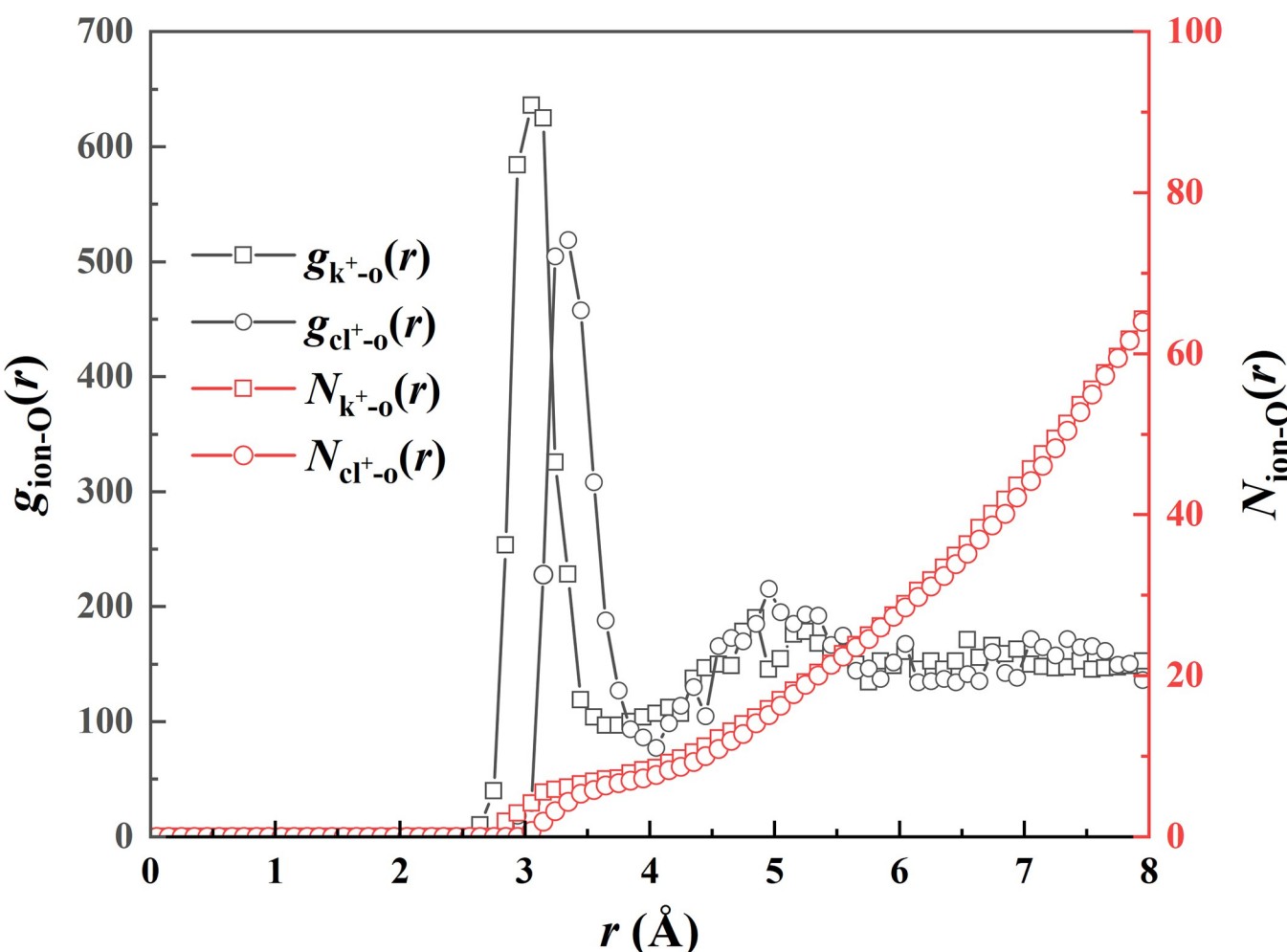

**Fig 8. Radial distribution functions $g_{\text{ion-O}}(r)$ together with their integrals $N_{\text{ion-O}}(r)$ corresponding to generation of secondary droplets at $t = 187$ ps.**

indicates that, when electrocoalescence process produces some tiny daughter droplets, the corresponding hydration number for $K^+$ and $Cl^-$ are 6.32 and 7.20. Therefore, water molecules show a relative weak constraint force for $K^+$ compared with that for $Cl^-$, which determines molecular cluster with dissolved $K^+$ is readily ejected from the primary droplet to form a larger secondary droplet. After that, the coalescing droplet is stretched to the line-like droplet, and the breakup of coalescing droplet occurs (due to the fact of the ions exchange), resulting in the non-coalescence dynamics at $t = 215$ ps. Thus, the critical electric field for triggering the non-coalescence dynamics is $E_{cn} = 0.51$ V nm$^{-1}$. As a result, there are two critical electric fields in terms of the electrocoalescence process: one is to induce the chain configuration at a small simulated domain ($E_{cc}$), while the other one is to induce the non-coalescence dynamics at a sufficiently large domain ($E_{cn}$). It is worth pointing out that the proton conductivity in liquid water at the nanoscale is significantly enhanced under the action of an external electric field [38]. Cassone provided a deep insights into the phenomenon of the enhanced proton conductivity and was ascribed to nuclear quantum effects [38]. For process of the filed-induced coalescence, the electrostatic force between two adjacent nanodroplets may be increased, which may directly lead to diverse results compared with that of electrocoalescence for macro-droplets: (I) acceerative droplets coalescence, and reduction of critical electric field for (II) chain configuration, $E_{cc}$, and (III) non-coalescence process.

## 3.3 Suppression of water chain or non-coalescence dynamics

Inspired by the previous studies, we perform the pulsed DC electric field to overcome the disadvantage of applying DC electric field, to explore the detailed dynamics and the corresponding mechanics behind it. The applying pulsed DC electric field with $E_{cc} = 0.4$ V nm$^{-1}$ and $E_{cn} = 0.51$ V nm$^{-1}$ are selected to investigate the dynamic electro-coalescence behaviors, as shown in **Fig 9**. Notice that, for the selected pulsed DC electric field, the pulse period is maintained at a constant value of 120 ps with the duty ratio of 0.5, i.e. the pulsed duration is 60 ps equaling to that for the pulsed interval. For the coalescence case of a small domain in **Fig 9A**, three droplets are elongated to prolate spheroid during the first pulsed duration, and the coalescence not yet occurred (60 ps). During the pulsed interval, the removal of the electric field makes water droplets recover the spherical shape by the surface force. Droplets contact and coalesce into a bigger coalescing one occurring in the second pulsed, and then such a coalescing droplet undergoes the periodical deformation, being in an elongated one within the pulsed duration whereas a spherical one within the pulsed interval, till the simulated end. Under a higher electric field of 0.51 V nm$^{-1}$, the separating droplets contact with each other within the first pulsed duration, as shown at $t = 53$ ps in **Fig 9B**. The similar dynamics of the coalescing droplet as that for the small domain are observed with the periodic deformation. We introduce the equivalent electric field strength ($E_{e\text{-}e\text{-}f}$) to explore the mechanics of the suppressing chain formation or non-coalescence dynamics, which is the product of RMS value and $E_{max}$. And, the RMS value is defined as the square root of the average of the squares of a set of numbers or quantities. According to the definition, the values of the small and large simulated domains can be calculated as $E_{e\text{-}e\text{-}f} = 0.28$ V nm$^{-1}$ and 0.36 V nm$^{-1}$. So, applying a pulsed DC electric field can reduce $E_{e\text{-}e\text{-}f}$ ($E_{e\text{-}e\text{-}f} = 0.4$ and 0.51 V nm$^{-1}$ for the DC electric field), so as to successful suppression of the chain configuration and the non-coalescence dynamics. We then gradually increase electric fields to explore the critical field strength of the pulsed DC field for inducing the chain configuration and the non-coalescence dynamics, and their values are confirmed to be 0.55 and 0.71 V nm$^{-1}$. Interestingly, we find that the equivalent electric field strength of such cases is exactly equal to that of the DC electric field. Therefore, we confirm that the adverse condition of applying the DC electric field, including the chain configuration and the non-

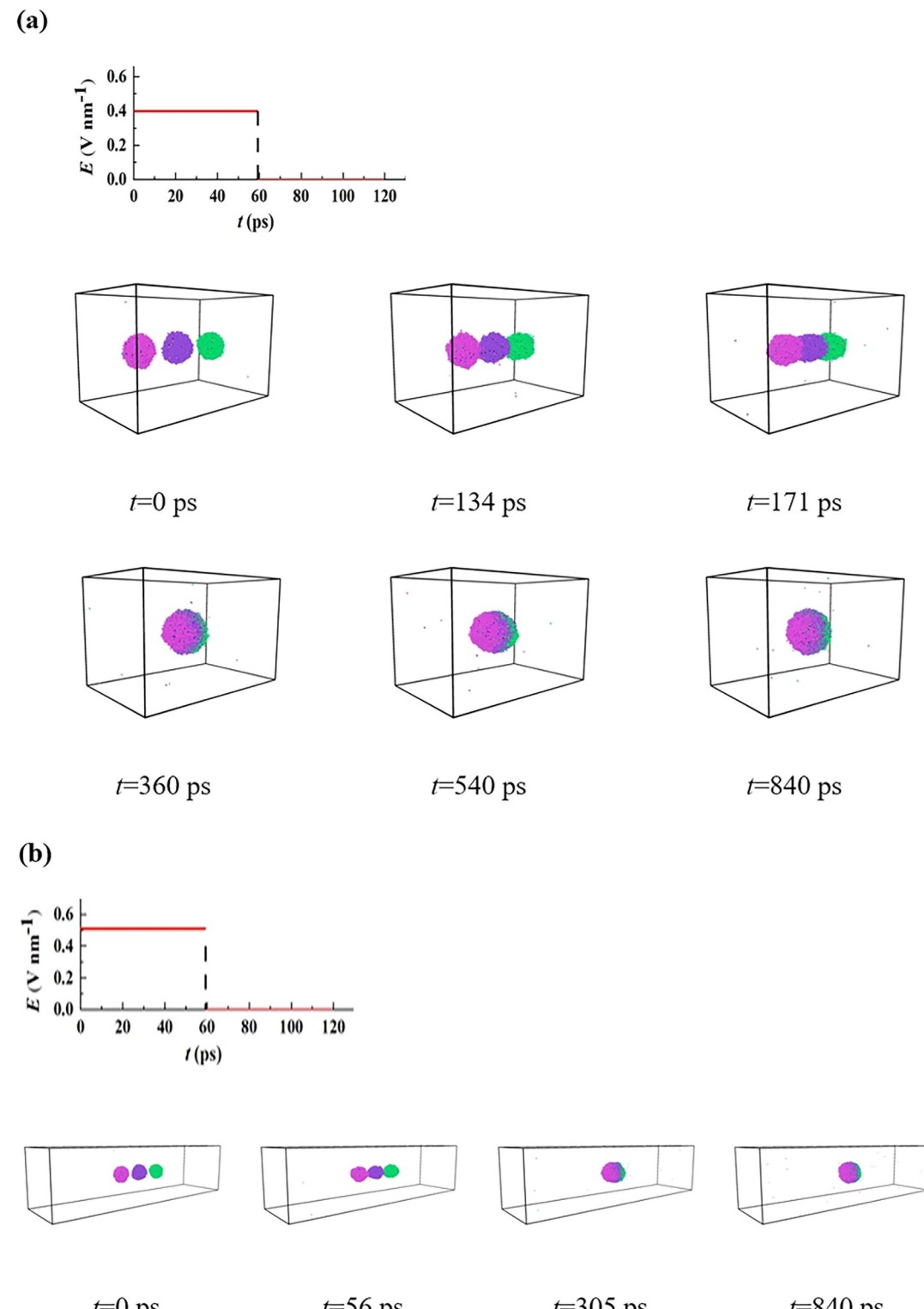

**Fig 9.** Applying pulsed DC electric field of (a) 0.4 V nm$^{-1}$ to small simulated domain, and (b) 0.7 V nm$^{-1}$ to large simulated domain to induce the droplets coalescence.

coalescence dynamics, can be surmounted if the applying electric field is replaced by the pulsed DC electric field. And, the critical electric fields, $E_{cc}$ and $E_{cn}$, can be greatly increased which only depends on the value of the equivalent electric field strength, $E_{e-e-f}$.

## 4. Conclusions

In summary, we investigate the dynamic electro-coalescence process of three droplet systems under constant DC electric fields. We show that the coalescence dynamics are almost symmetrical when separating gap thickness possesses the same value. Whereas the asymmetrical dynamics can be observed with the different gap thicknesses separating droplets, that is, the coalescence of droplets with a small distance should be prior to that for the large separating gap thickness. We find that there is a critical electric field ($E_{cc}$) above which the coalescing droplet loses its ability to control the droplet shape, and the droplet is elongated under the action of $F_i$ together with $F_e$ (stemming from the adjacent domain) to form the stable chain configuration. Owing to the significant reduction of $F_e$, the occurrence of the complete coalescence under the same electric field is possible with the increasing simulated domain. Further increasing electric field strength would promote another different dynamics. We observe that some secondary droplets first eject from the coalescing droplet (partial coalescence) and then the liquid bridge may be destroyed by the electric field force leading to non-coalescence dynamics. Using the pulsed DC electric field, both chain configuration and the non-coalescence procedure can be effectively avoided. This is due to the fact that the pulsed DC electric field can greatly reduce the equivalent electric field strength ($E_{e-e-f}$) to further prevent adverse dynamics. However, the chain configuration or non-coalescence is inevitable when the $E_{e-e-f}$ of the pulsed electric field is increased to the same value as that for the DC electric field.

## Supporting information

**S1 Data.**
(ZIP)

## Author Contributions

**Data curation:** Liwei Sun, Yanhui Chen.

**Formal analysis:** Jian Wang.

**Funding acquisition:** Yanhui Chen.

**Investigation:** Liwei Sun, Jian Wang.

**Methodology:** Liwei Sun, Jian Wang.

**Project administration:** Yanhui Chen.

**Software:** Liwei Sun, Jian Wang.

**Supervision:** Jian Wang, Yanhui Chen.

**Validation:** Liwei Sun, Jian Wang, Yanhui Chen.

**Visualization:** Liwei Sun, Jian Wang.

**Writing – original draft:** Liwei Sun.

**Writing – review & editing:** Yanhui Chen.

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
