## [Decision Letter · Decision Letter 0]

2 Jan 2024

PONE-D-23-42147Coalescence of multiple droplets induced by a constant DC electric fieldPLOS ONE

Dear Dr. Chen,

Thank you for submitting your manuscript to PLOS ONE. After careful consideration, we feel that it has merit but does not fully meet PLOS ONE’s publication criteria as it currently stands. Therefore, we invite you to submit a revised version of the manuscript that addresses the points raised during the review process.

We look forward to receiving your revised manuscript.

Kind regards,

Jianhui Liu

Academic Editor

PLOS ONE

Journal Requirements:

Reviewers' comments:

Reviewer's Responses to Questions

**Comments to the Author**

1. Is the manuscript technically sound, and do the data support the conclusions?

Reviewer #1: Yes

Reviewer #2: Partly

2. Has the statistical analysis been performed appropriately and rigorously? 

Reviewer #1: Yes

Reviewer #2: Yes

3. Have the authors made all data underlying the findings in their manuscript fully available?

Reviewer #1: Yes

Reviewer #2: No

4. Is the manuscript presented in an intelligible fashion and written in standard English?

Reviewer #1: Yes

Reviewer #2: No

5. Review Comments to the Author

Reviewer #1: In this paper, the molecular dynamics simulation method was used to study the coalescence behavior of multiple nanodroplets under an electric field. This is an interesting work, and the research results obtained have a certain guiding significance for the electrical control of droplets. However, the following problems still need to be solved:

1. Please add validation of the model, which is essential and also supports the credibility of subsequent research.

2. The ion content has an important effect on the electrocoalescence behavior of nanodroplets. What is the basis for determining the ion concentration in this paper? Will different ion contents affect the conclusions of this paper?

3. What is the specific number of droplets water molecules in the model? Is it determined based on density or some other principle?

4. The environment of the droplets in this model is a vacuum, but in practical applications, the environment of the droplets is generally a gas or insoluble liquid medium. How can the author ensure that the electric coalescence behavior of the droplets in a vacuum is scientific and reliable in guiding the real situation? Currently, most of the molecular dynamics studies on the electrocoalescence of nanodroplets have been carried out in boxes with continuous phases of gas or liquid molecules.

5. It is mentioned in this paper that different droplet dynamic behaviors have been observed in different box sizes under the same working conditions. Is this the cause of model construction or electrohydrodynamic? If the constructed PBC box is inappropriate, the author should determine a box size that best corresponds to the experimental phenomenon or practical application. If there are differences in the electrohydrodynamic factors of droplets under different box sizes, the authors should clarify the mechanism.

Reviewer #2: The Authors present an interesting investigation on the capabilities hold by DC electric fields in producing and sustaining electro-coalescence. The topic is very actual and interesting and most of the data the Authors report are sound. However, I have several concerns regarding the completeness of the work, the way in which it has been conducted and written, that force me to suggest to propose major revisions before re-considering the paper for eventual acceptance. Please find in the following the most important issues.

*) Abstract can be improved to a large extent. The reader approaching the abstract for the 1st time cannot understand part of it. As an example, both in the abstract and elsewhere in the main text the definition of the "equivalent electric field strength" remains completely obscure and elusive.

*) Introduction: This section is very poorly written and the English level is very scarce, in my opinion. I suggest extensive re-writing of this section. Moreover, the bibliographic apparatus presented in this Section is definitely incomplete. In the last decade, indeed, several important computational article dealing with water under electric fields have been reported in the literature [JCP 158, 184501 (2023); JCP 147, 031102 (2017); JCP 150, 074505 (2019)], just to cite a few. None of these important works is cited in the manuscript.

*) Methods: In this section the Authors state that "the dynamic electrocoalescence of three nanodroplet systems under a constant electric field is first investigated via MD simulations." However, they present the results of only one single system composed of three nanodroplets. The initial sentence is completely misleading.

*) Methods: The Authors use the non-polarisable SPC/E water model in presence of strongly polarizing electric fields. Such an intrinsic and crucial limitation must be stressed both in the Methods section and throughout the manuscript as well. Of course, neglecting these important effects might limit the predictability level of the presented findings.

*) Methods: The Authors state that they use "charged nanodroplets". However, they never specify how these are charged since they are composed of only neutral water and KCl (globally neutral) species. Please specify how these droplets result to be charged and how much charged they are! This is essential to figure out the macroscopic electrostatic force exerted by the field on them.

*) In Figure 5 all field strengths are in the order of 1-5 V/nm whereas in the text the Authors report one order of magnitude fields (0.1-0.5 V/nm). Please PAY ATTENTION on the strength of the field! One order of magnitude of difference may lead (and in general leads!) to completely different responses of molecular systems.

*) Related to the previous point is the fact that a large fraction of the water molecules should be re-oriented by the field [JCPL 11, 8983 (2020); PCCP 21, 21205 (2019)]. Could the Authors comment more on this aspect and/or determine the amount of re-oriented water molecules? This is very important to understand how the field is capable of, e.g., increasing the total dipole moment inside the nanodroplets.

*) General comment: the results the Author report are interesting but rather qualitative and descriptive. Maybe, they could exploit the revision process to go deeper with some more quantitative analysis that would improve the quality of the findings reported in this work.

6. PLOS authors have the option to publish the peer review history of their article (what does this mean?). If published, this will include your full peer review and any attached files.

Reviewer #1: No

Reviewer #2: No

---

## [Author Response · Author response to Decision Letter 0]

29 Jan 2024

We have respond to specific reviewer and editor comments upload as a separate file.

---

## [Decision Letter · Decision Letter 1]

9 Feb 2024

PONE-D-23-42147R1Coalescence of multiple droplets induced by a constant DC electric fieldPLOS ONE

Dear Dr. Chen,

Thank you for submitting your manuscript to PLOS ONE. After careful consideration, we feel that it has merit but does not fully meet PLOS ONE’s publication criteria as it currently stands. Therefore, we invite you to submit a revised version of the manuscript that addresses the points raised during the review process.

We look forward to receiving your revised manuscript.

Kind regards,

Jianhui Liu

Academic Editor

PLOS ONE

Journal Requirements:

Reviewers' comments:

Reviewer's Responses to Questions

**Comments to the Author**

1. If the authors have adequately addressed your comments raised in a previous round of review and you feel that this manuscript is now acceptable for publication, you may indicate that here to bypass the “Comments to the Author” section, enter your conflict of interest statement in the “Confidential to Editor” section, and submit your "Accept" recommendation.

Reviewer #1: All comments have been addressed

Reviewer #2: (No Response)

2. Is the manuscript technically sound, and do the data support the conclusions?

Reviewer #1: Yes

Reviewer #2: Yes

3. Has the statistical analysis been performed appropriately and rigorously? 

Reviewer #1: Yes

Reviewer #2: Yes

4. Have the authors made all data underlying the findings in their manuscript fully available?

Reviewer #1: Yes

Reviewer #2: (No Response)

5. Is the manuscript presented in an intelligible fashion and written in standard English?

Reviewer #1: Yes

Reviewer #2: (No Response)

6. Review Comments to the Author

Reviewer #1: The authors have answered the questionsi and edit this paper. The data support the article. I think this can be published.

Reviewer #2: The Authors have put some effort in addressing most, but not all, of the points I've raised during the previous round of review.

First of all, the English level of the manuscript remained relatively poor, even though some effort has been done in improving its readability.

On the scientific perspective, I believe that the Authors should consider more seriously to include some relevant bilbiography I suggested in the

previous round where the effects carried by electric fields in water are treated at the quantum level (see, e.g., [JCPL 11, 8983 (2020)]). Unfortunately, the Authors didn't take advantage of the revision process to include this aspect, which is tightly related to some of the results the Authors report here. In fact, from that work turns out the timescales and the dynamics associated with the re-orientation of the water dipoles under the field effect while considering the water molecules at the electronic and nuclear quantum level. Unfortunately, the Authors completely skip my suggestion.

Once those two aspects will be taken into account more seriously, it will be my pleasure to re-consider the paper for publication.

7. PLOS authors have the option to publish the peer review history of their article (what does this mean?). If published, this will include your full peer review and any attached files.

Reviewer #1: No

Reviewer #2: No

---

## [Author Response · Author response to Decision Letter 1]

4 Mar 2024

The respond to reviewer and editor comments have uploaded as a separate file.

---

## [Decision Letter · Decision Letter 2]

7 Mar 2024

Coalescence of multiple droplets induced by a constant DC electric field

PONE-D-23-42147R2

Dear Dr. Chen,

We’re pleased to inform you that your manuscript has been judged scientifically suitable for publication and will be formally accepted for publication once it meets all outstanding technical requirements.

Kind regards,

Jianhui Liu

Academic Editor

PLOS ONE

Additional Editor Comments (optional):

Reviewers' comments:

Reviewer's Responses to Questions

**Comments to the Author**

1. If the authors have adequately addressed your comments raised in a previous round of review and you feel that this manuscript is now acceptable for publication, you may indicate that here to bypass the “Comments to the Author” section, enter your conflict of interest statement in the “Confidential to Editor” section, and submit your "Accept" recommendation.

Reviewer #2: All comments have been addressed

2. Is the manuscript technically sound, and do the data support the conclusions?

Reviewer #2: Yes

3. Has the statistical analysis been performed appropriately and rigorously? 

Reviewer #2: Yes

4. Have the authors made all data underlying the findings in their manuscript fully available?

Reviewer #2: Yes

5. Is the manuscript presented in an intelligible fashion and written in standard English?

Reviewer #2: Yes

6. Review Comments to the Author

Reviewer #2: The paper can now be accepted because the Authors made a great effort in revising it. Also, the English level has been improved.

7. PLOS authors have the option to publish the peer review history of their article (what does this mean?). If published, this will include your full peer review and any attached files.

Reviewer #2: No

---

## [Editor Report · Acceptance letter]

30 Mar 2024

PONE-D-23-42147R2 

PLOS ONE

Dear Dr. Chen, 

I'm pleased to inform you that your manuscript has been deemed suitable for publication in PLOS ONE. Congratulations! Your manuscript is now being handed over to our production team.

Kind regards, 

on behalf of

Dr. Jianhui Liu 

Academic Editor

PLOS ONE